# A Latent Source Model for Nonparametric Time Series Classification

**George H. Chen**
MIT
georgehc@mit.edu

**Stanislav Nikolov**
Twitter
snikolov@twitter.com

**Devavrat Shah**
MIT
devavrat@mit.edu

## Abstract

For classifying time series, a nearest-neighbor approach is widely used in practice with performance often competitive with or better than more elaborate methods such as neural networks, decision trees, and support vector machines. We develop theoretical justification for the effectiveness of nearest-neighbor-like classification of time series. Our guiding hypothesis is that in many applications, such as forecasting which topics will become trends on Twitter, there aren't actually that many prototypical time series to begin with, relative to the number of time series we have access to, e.g., topics become trends on Twitter only in a few distinct manners whereas we can collect massive amounts of Twitter data. To operationalize this hypothesis, we propose a *latent source model* for time series, which naturally leads to a "weighted majority voting" classification rule that can be approximated by a nearest-neighbor classifier. We establish nonasymptotic performance guarantees of both weighted majority voting and nearest-neighbor classification under our model accounting for how much of the time series we observe and the model complexity. Experimental results on synthetic data show weighted majority voting achieving the same misclassification rate as nearest-neighbor classification while observing less of the time series. We then use weighted majority to forecast which news topics on Twitter become trends, where we are able to detect such "trending topics" in advance of Twitter 79% of the time, with a mean early advantage of 1 hour and 26 minutes, a true positive rate of 95%, and a false positive rate of 4%.

## 1 Introduction

Recent years have seen an explosion in the availability of time series data related to virtually every human endeavor — data that demands to be analyzed and turned into valuable insights. A key recurring task in mining this data is being able to classify a time series. As a running example used throughout this paper, consider a time series that tracks how much activity there is for a particular news topic on Twitter. Given this time series up to present time, we ask "will this news topic go viral?" Borrowing Twitter's terminology, we label the time series a "trend" and call its corresponding news topic a *trending topic* if the news topic goes viral; otherwise, the time series has label "not trend". We seek to forecast whether a news topic will become a trend *before* it is declared a trend (or not) by Twitter, amounting to a binary classification problem. Importantly, we skirt the discussion of what makes a topic considered trending as this is irrelevant to our mathematical development.[1] Furthermore, we remark that handling the case where a single time series can have different labels at different times is beyond the scope of this paper.

Numerous standard classification methods have been tailored to classify time series, yet a simple nearest-neighbor approach is hard to beat in terms of classification performance on a variety of datasets [20], with results competitive to or better than various other more elaborate methods such as neural networks [15], decision trees [16], and support vector machines [19]. More recently, researchers have examined which distance to use with nearest-neighbor classification [2, 7, 18] or how to boost classification performance by applying different transformations to the time series before using nearest-neighbor classification [1]. These existing results are mostly experimental, lacking theoretical justification for both when nearest-neighbor-like time series classifiers should be expected to perform well and how well.

If we don't confine ourselves to classifying time series, then as the amount of data tends to infinity, nearest-neighbor classification has been shown to achieve a probability of error that is at worst twice the Bayes error rate, and when considering the nearest $k$ neighbors with $k$ allowed to grow with the amount of data, then the error rate approaches the Bayes error rate [5]. However, rather than examining the asymptotic case where the amount of data goes to infinity, we instead pursue *nonasymptotic* performance guarantees in terms of how large of a training dataset we have and how much we observe of the time series to be classified. To arrive at these nonasymptotic guarantees, we impose a low-complexity structure on time series.

**Our contributions.** We present a model for which nearest-neighbor-like classification performs well by operationalizing the following hypothesis: In many time series applications, there are only a small number of prototypical time series relative to the number of time series we can collect. For example, posts on Twitter are generated by humans, who are often behaviorally predictable in aggregate. This suggests that topics they post about only become trends on Twitter in a few distinct manners, yet we have at our disposal enormous volumes of Twitter data. In this context, we present a novel *latent source model*: time series are generated from a small collection of $m$ unknown latent sources, each having one of two labels, say "trend" or "not trend". Our model's maximum a posteriori (MAP) time series classifier can be approximated by weighted majority voting, which compares the time series to be classified with each of the time series in the labeled training data. Each training time series casts a weighted vote in favor of its ground truth label, with the weight depending on how similar the time series being classified is to the training example. The final classification is "trend" or "not trend" depending on which label has the higher overall vote. The voting is nonparametric in that it does not learn parameters for a model and is driven entirely by the training data. The unknown latent sources are never estimated; the training data serve as a proxy for these latent sources. Weighted majority voting itself can be approximated by a nearest-neighbor classifier, which we also analyze.

Under our model, we show sufficient conditions so that if we have $n = \Theta(m \log \frac{m}{\delta})$ time series in our training data, then weighted majority voting and nearest-neighbor classification correctly classify a new time series with probability at least $1 - \delta$ after observing its first $\Omega(\log \frac{m}{\delta})$ time steps. As our analysis accounts for how much of the time series we observe, our results readily apply to the "online" setting in which a time series is to be classified while it streams in (as is the case for forecasting trending topics) as well as the "offline" setting where we have access to the entire time series. Also, while our analysis yields matching error upper bounds for the two classifiers, experimental results on synthetic data suggests that weighted majority voting outperforms nearest-neighbor classification early on when we observe very little of the time series to be classified. Meanwhile, a specific instantiation of our model leads to a spherical Gaussian mixture model, where the latent sources are Gaussian mixture components. We show that existing performance guarantees on learning spherical Gaussian mixture models [6, 10, 17] require more stringent conditions than what our results need, suggesting that learning the latent sources is overkill if the goal is classification.

Lastly, we apply weighted majority voting to forecasting trending topics on Twitter. We emphasize that our goal is *precognition* of trends: predicting whether a topic is going to be a trend before it is actually declared to be a trend by Twitter or, in theory, any other third party that we can collect ground truth labels from. Existing work that identify trends on Twitter [3, 4, 13] instead, as part of their trend detection, define models for what trends are, which we do not do, nor do we assume we have access to such definitions. (The same could be said of previous work on novel document detection on Twitter [11, 12].) In our experiments, weighted majority voting is able to predict whether a topic will be a trend in advance of Twitter 79% of the time, with a mean early advantage of 1 hour and 26 minutes, a true positive rate of 95%, and a false positive rate of 4%. We empirically find that the Twitter activity of a news topic that becomes a trend tends to follow one of a finite number of patterns, which could be thought of as latent sources.

**Outline.** Weighted majority voting and nearest-neighbor classification for time series are presented in Section 2. We provide our latent source model and theoretical performance guarantees of weighted majority voting and nearest-neighbor classification under this model in Section 3. Experimental results for synthetic data and forecasting trending topics on Twitter are in Section 4.

## 2 Weighted Majority Voting and Nearest-Neighbor Classification

Given a time-series[2] $s : \mathbb{Z} \to \mathbb{R}$, we want to classify it as having either label $+1$ ("trend") or $-1$ ("not trend"). To do so, we have access to labeled training data $\mathcal{R}_+$ and $\mathcal{R}_-$, which denote the sets of all training time series with labels $+1$ and $-1$ respectively.

**Weighted majority voting.** Each positively-labeled example $r \in \mathcal{R}_+$ casts a weighted vote $e^{-\gamma d^{(T)}(r,s)}$ for whether time series $s$ has label $+1$, where $d^{(T)}(r,s)$ is some measure of similarity between the two time series $r$ and $s$, superscript $(T)$ indicates that we are only allowed to look at the first $T$ time steps (i.e., time steps $1, 2, \dots, T$) of $s$ (but we're allowed to look outside of these time steps for the training time series $r$), and constant $\gamma \geq 0$ is a scaling parameter that determines the "sphere of influence" of each example. Similarly, each negatively-labeled example in $\mathcal{R}_-$ also casts a weighted vote for whether time series $s$ has label $-1$.

The similarity measure $d^{(T)}(r,s)$ could, for example, be squared Euclidean distance: $d^{(T)}(r,s) = \sum_{t=1}^{T}(r(t) - s(t))^2 \triangleq \|r - s\|_T^2$. However, this similarity measure only looks at the first $T$ time steps of training time series $r$. Since time series in our training data are known, we need not restrict our attention to their first $T$ time steps. Thus, we use the following similarity measure:

$$d^{(T)}(r,s) = \min_{\Delta \in \{-\Delta_{\max}, \dots, 0, \dots, \Delta_{\max}\}} \sum_{t=1}^{T}(r(t+\Delta) - s(t))^2 = \min_{\Delta \in \{-\Delta_{\max}, \dots, 0, \dots, \Delta_{\max}\}} \|r * \Delta - s\|_T^2, \tag{1}$$

where we minimize over integer time shifts with a pre-specified maximum allowed shift $\Delta_{\max} \geq 0$. Here, we have used $q * \Delta$ to denote time series $q$ advanced by $\Delta$ time steps, i.e., $(q * \Delta)(t) = q(t+\Delta)$.

Finally, we sum up all of the weighted $+1$ votes and then all of the weighted $-1$ votes. The label with the majority of overall weighted votes is declared as the label for $s$:

$$\widehat{L}^{(T)}(s;\gamma) = \begin{cases} +1 & \text{if } \sum_{r \in \mathcal{R}_+} e^{-\gamma d^{(T)}(r,s)} \geq \sum_{r \in \mathcal{R}_-} e^{-\gamma d^{(T)}(r,s)}, \\ -1 & \text{otherwise.} \end{cases} \tag{2}$$

Using a larger time window size $T$ corresponds to waiting longer before we make a prediction. We need to trade off how long we wait and how accurate we want our prediction. Note that $k$-nearest-neighbor classification corresponds to only considering the $k$ nearest neighbors of $s$ among all training time series; all other votes are set to 0. With $k = 1$, we obtain the following classifier:

**Nearest-neighbor classifier.** Let $\widehat{r} = \arg\min_{r \in \mathcal{R}_+ \cup \mathcal{R}_-} d^{(T)}(r,s)$ be the nearest neighbor of $s$. Then we declare the label for $s$ to be:

$$\widehat{L}_{NN}^{(T)}(s) = \begin{cases} +1 & \text{if } \widehat{r} \in \mathcal{R}_+, \\ -1 & \text{if } \widehat{r} \in \mathcal{R}_-. \end{cases} \tag{3}$$

## 3 A Latent Source Model and Theoretical Guarantees

We assume there to be $m$ unknown latent sources (time series) that generate observed time series. Let $\mathcal{V}$ denote the set of all such latent sources; each latent source $v : \mathbb{Z} \to \mathbb{R}$ in $\mathcal{V}$ has a true label $+1$ or $-1$. Let $\mathcal{V}_+ \subset \mathcal{V}$ be the set of latent sources with label $+1$, and $\mathcal{V}_- \subset \mathcal{V}$ be the set of those with label $-1$. The observed time series are generated from latent sources as follows:

1. Sample latent source $V$ from $\mathcal{V}$ uniformly at random.[3] Let $L \in \{\pm 1\}$ be the label of $V$.

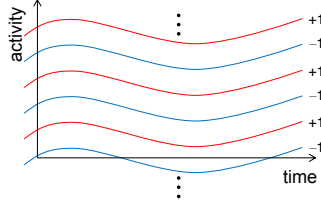

Figure 1: Example of latent sources superimposed, where each latent source is shifted vertically in amplitude such that every other latent source has label $+1$ and the rest have label $-1$.

2. Sample integer time shift $\Delta$ uniformly from $\{0, 1, \ldots, \Delta_{\max}\}$.

3. Output time series $S : \mathbb{Z} \to \mathbb{R}$ to be latent source $V$ advanced by $\Delta$ time steps, followed by adding noise signal $E : \mathbb{Z} \to \mathbb{R}$, i.e., $S(t) = V(t + \Delta) + E(t)$. The label associated with the generated time series $S$ is the same as that of $V$, i.e., $L$. Entries of noise $E$ are i.i.d. zero-mean sub-Gaussian with parameter $\sigma$, which means that for any time index $t$,

$$\mathbb{E}[\exp(\lambda E(t))] \leq \exp\left(\frac{1}{2}\lambda^2 \sigma^2\right) \qquad \text{for all } \lambda \in \mathbb{R}. \tag{4}$$

The family of sub-Gaussian distributions includes a variety of distributions, such as a zero-mean Gaussian with standard deviation $\sigma$ and a uniform distribution over $[-\sigma, \sigma]$.

The above generative process defines our latent source model. Importantly, we make no assumptions about the structure of the latent sources. For instance, the latent sources could be tiled as shown in Figure 1, where they are evenly separated vertically and alternate between the two different classes $+1$ and $-1$. With a parametric model like a $k$-component Gaussian mixture model, estimating these latent sources could be problematic. For example, if we take any two adjacent latent sources with label $+1$ and cluster them, then this cluster could be confused with the latent source having label $-1$ that is sandwiched in between. Noise only complicates estimating the latent sources. In this example, the $k$-component Gaussian mixture model needed for label $+1$ would require $k$ to be the exact number of latent sources with label $+1$, which is unknown. In general, the number of samples we need from a Gaussian mixture mixture model to estimate the mixture component means is exponential in the number of mixture components [14]. As we discuss next, for classification, we sidestep learning the latent sources altogether, instead using training data as a proxy for latent sources. At the end of this section, we compare our sample complexity for classification versus some existing sample complexities for learning Gaussian mixture models.

**Classification.** If we knew the latent sources and if noise entries $E(t)$ were i.i.d. $\mathcal{N}(0, \frac{1}{2\gamma})$ across $t$, then the *maximum a posteriori* (MAP) estimate for label $L$ given an observed time series $S = s$ is

$$\widehat{L}_{\text{MAP}}^{(T)}(s; \gamma) = \begin{cases} +1 & \text{if } \Lambda_{\text{MAP}}^{(T)}(s; \gamma) \geq 1, \\ -1 & \text{otherwise,} \end{cases} \tag{5}$$

where

$$\Lambda_{\text{MAP}}^{(T)}(s; \gamma) \triangleq \frac{\sum_{v_+ \in \mathcal{V}_+} \sum_{\Delta_+ \in \mathcal{D}_+} \exp\left(-\gamma \|v_+ * \Delta_+ - s\|_T^2\right)}{\sum_{v_- \in \mathcal{V}_-} \sum_{\Delta_- \in \mathcal{D}_+} \exp\left(-\gamma \|v_- * \Delta_- - s\|_T^2\right)}, \tag{6}$$

and $\mathcal{D}_+ \triangleq \{0, \ldots, \Delta_{\max}\}$.

However, we do not know the latent sources, nor do we know if the noise is i.i.d. Gaussian. We assume that we have access to training data as given in Section 2. We make a further assumption that the training data were sampled from the latent source model and that we have $n$ different training time series. Denote $\mathcal{D} \triangleq \{-\Delta_{\max}, \ldots, 0, \ldots, \Delta_{\max}\}$. Then we approximate the MAP classifier by using training data as a proxy for the latent sources. Specifically, we take ratio (6), replace the inner sum by a minimum in the exponent, replace $\mathcal{V}_+$ and $\mathcal{V}_-$ by $\mathcal{R}_+$ and $\mathcal{R}_-$, and replace $\mathcal{D}_+$ by $\mathcal{D}$ to obtain the ratio:

$$\Lambda^{(T)}(s; \gamma) \triangleq \frac{\sum_{r_+ \in \mathcal{R}_+} \exp\left(-\gamma\left(\min_{\Delta_+ \in \mathcal{D}} \|r_+ * \Delta_+ - s\|_T^2\right)\right)}{\sum_{r_- \in \mathcal{R}_-} \exp\left(-\gamma\left(\min_{\Delta_- \in \mathcal{D}} \|r_- * \Delta_- - s\|_T^2\right)\right)}. \tag{7}$$

Plugging $\Lambda^{(T)}$ in place of $\Lambda^{(T)}_{\mathrm{MAP}}$ in classification rule (5) yields the weighted majority voting rule (2). Note that weighted majority voting could be interpreted as a *smoothed* nearest-neighbor approximation whereby we only consider the time-shifted version of each example time series that is closest to the observed time series $s$. If we didn't replace the summations over time shifts with minimums in the exponent, then we have a kernel density estimate in the numerator and in the denominator [9, Chapter 7] (where the kernel is Gaussian) and our main theoretical result for weighted majority voting to follow would still hold using the same proof.[4]

Lastly, applications may call for trading off true and false positive rates. We can do this by generalizing decision rule (5) to declare the label of $s$ to be $+1$ if $\Lambda^{(T)}(s, \gamma) \geq \theta$ and vary parameter $\theta > 0$. The resulting decision rule, which we refer to as *generalized weighted majority voting*, is thus:

$$\widehat{L}^{(T)}_{\theta}(s; \gamma) = \begin{cases} +1 & \text{if } \Lambda^{(T)}(s, \gamma) \geq \theta, \\ -1 & \text{otherwise,} \end{cases} \tag{8}$$

where setting $\theta = 1$ recovers the usual weighted majority voting (2). This modification to the classifier can be thought of as adjusting the priors on the relative sizes of the two classes. Our theoretical results to follow actually cover this more general case rather than only that of $\theta = 1$.

**Theoretical guarantees.** We now present the main theoretical results of this paper which identify sufficient conditions under which generalized weighted majority voting (8) and nearest-neighbor classification (3) can classify a time series correctly with high probability, accounting for the size of the training dataset and how much we observe of the time series to be classified. First, we define the "gap" between $\mathcal{R}_+$ and $\mathcal{R}_-$ restricted to time length $T$ and with maximum time shift $\Delta_{\max}$ as:

$$G^{(T)}(\mathcal{R}_+, \mathcal{R}_-, \Delta_{\max}) \triangleq \min_{\substack{r_+ \in \mathcal{R}_+, r_- \in \mathcal{R}_-, \\ \Delta_+, \Delta_- \in \mathcal{D}}} \|r_+ * \Delta_+ - r_- * \Delta_-\|^2_T. \tag{9}$$

This quantity measures how far apart the two different classes are if we only look at length-$T$ chunks of each time series and allow all shifts of at most $\Delta_{\max}$ time steps in either direction.

Our first main result is stated below. We defer proofs to the longer version of this paper.

**Theorem 1.** *(Performance guarantee for generalized weighted majority voting) Let $m_+ = |\mathcal{V}_+|$ be the number of latent sources with label $+1$, and $m_- = |\mathcal{V}_-| = m - m_+$ be the number of latent sources with label $-1$. For any $\beta > 1$, under the latent source model with $n > \beta m \log m$ time series in the training data, the probability of misclassifying time series $S$ with label $L$ using generalized weighted majority voting $\widehat{L}^{(T)}_{\theta}(\cdot; \gamma)$ satisfies the bound*

$$\mathbb{P}(\widehat{L}^{(T)}_{\theta}(S; \gamma) \neq L)$$
$$\leq \left( \frac{\theta m_+}{m} + \frac{m_-}{\theta m} \right)(2\Delta_{\max} + 1)n \exp\left( -(\gamma - 4\sigma^2\gamma^2)G^{(T)}(\mathcal{R}_+, \mathcal{R}_-, \Delta_{\max}) \right) + m^{-\beta+1}. \tag{10}$$

An immediate consequence is that given error tolerance $\delta \in (0, 1)$ and with choice $\gamma \in (0, \frac{1}{4\sigma^2})$, then upper bound (10) is at most $\delta$ (by having each of the two terms on the right-hand side be $\leq \frac{\delta}{2}$) if $n > m \log \frac{2m}{\delta}$ (i.e., $\beta = 1 + \log \frac{2}{\delta} / \log m$), and

$$G^{(T)}(\mathcal{R}_+, \mathcal{R}_-, \Delta_{\max}) \geq \frac{\log(\frac{\theta m_+}{m} + \frac{m_-}{\theta m}) + \log(2\Delta_{\max} + 1) + \log n + \log \frac{2}{\delta}}{\gamma - 4\sigma^2\gamma^2}. \tag{11}$$

This means that if we have access to a large enough pool of labeled time series, i.e., the pool has $\Omega(m \log \frac{m}{\delta})$ time series, then we can subsample $n = \Theta(m \log \frac{m}{\delta})$ of them to use as training data. Then with choice $\gamma = \frac{1}{8\sigma^2}$, generalized weighted majority voting (8) correctly classifies a new time series $S$ with probability at least $1 - \delta$ if

$$G^{(T)}(\mathcal{R}_+, \mathcal{R}_-, \Delta_{\max}) = \Omega\left( \sigma^2 \left( \log\left( \frac{\theta m_+}{m} + \frac{m_-}{\theta m} \right) + \log(2\Delta_{\max} + 1) + \log \frac{m}{\delta} \right) \right). \tag{12}$$

Thus, the gap between sets $\mathcal{R}_+$ and $\mathcal{R}_-$ needs to grow logarithmic in the number of latent sources $m$ in order for weighted majority voting to classify correctly with high probability. Assuming that the

original unknown latent sources are separated (otherwise, there is no hope to distinguish between the classes using any classifier) and the gap in the training data grows as $G^{(T)}(\mathcal{R}_+, \mathcal{R}_-, \Delta_{\max}) = \Omega(\sigma^2 T)$ (otherwise, the closest two training time series from opposite classes are within noise of each other), then observing the first $T = \Omega(\log(\theta + \frac{1}{\theta}) + \log(2\Delta_{\max} + 1) + \log \frac{m}{\delta})$ time steps from the time series is sufficient to classify it correctly with probability at least $1 - \delta$.

A similar result holds for the nearest-neighbor classifier (3).

**Theorem 2.** *(Performance guarantee for nearest-neighbor classification) For any $\beta > 1$, under the latent source model with $n > \beta m \log m$ time series in the training data, the probability of misclassifying time series $S$ with label $L$ using the nearest-neighbor classifier $\widehat{L}_{NN}^{(T)}(\cdot)$ satisfies the bound*

$$\mathbb{P}(\widehat{L}_{NN}^{(T)}(S) \neq L) \leq (2\Delta_{\max} + 1)n \exp\left(-\frac{1}{16\sigma^2}G^{(T)}(\mathcal{R}_+, \mathcal{R}_-, \Delta_{\max})\right) + m^{-\beta+1}. \quad (13)$$

Our generalized weighted majority voting bound (10) with $\theta = 1$ (corresponding to regular weighted majority voting) and $\gamma = \frac{1}{8\sigma^2}$ matches our nearest-neighbor classification bound, suggesting that the two methods have similar behavior when the gap grows with $T$. In practice, we find weighted majority voting to outperform nearest-neighbor classification when $T$ is small, and then as $T$ grows large, the two methods exhibit similar performance in agreement with our theoretical analysis. For small $T$, it could still be fairly likely that the nearest neighbor found has the wrong label, dooming the nearest-neighbor classifier to failure. Weighted majority voting, on the other hand, can recover from this situation as there may be enough correctly labeled training time series close by that contribute to a higher overall vote for the correct class. This robustness of weighted majority voting makes it favorable in the online setting where we want to make a prediction as early as possible.

**Sample complexity of learning the latent sources.** If we can estimate the latent sources accurately, then we could plug these estimates in place of the true latent sources in the MAP classifier and achieve classification performance close to optimal. If we restrict the noise to be Gaussian and assume $\Delta_{\max} = 0$, then the latent source model corresponds to a spherical Gaussian mixture model. We could learn such a model using Dasgupta and Schulman's modified EM algorithm [6]. Their theoretical guarantee depends on the true separation between the closest two latent sources, namely $G^{(T)*} \triangleq \min_{v,v' \in \mathcal{V} \text{ s.t. } v \neq v'} \|v - v'\|_2^2$, which needs to satisfy $G^{(T)*} \gg \sigma^2 \sqrt{T}$. Then with $n = \Omega(\max\{1, \frac{\sigma^2 T}{G^{(T)*}}\} m \log \frac{m}{\delta})$, $G^{(T)*} = \Omega(\sigma^2 \log \frac{m}{\varepsilon})$, and

$$T = \Omega\left(\max\left\{1, \frac{\sigma^4 T^2}{(G^{(T)*})^2}\right\} \log\left[\frac{m}{\delta} \max\left\{1, \frac{\sigma^4 T^2}{(G^{(T)*})^2}\right\}\right]\right), \quad (14)$$

their algorithm achieves, with probability at least $1 - \delta$, an additive $\varepsilon\sigma\sqrt{T}$ error (in Euclidean distance) close to optimal in estimating every latent source. In contrast, our result is in terms of gap $G^{(T)}(\mathcal{R}_+, \mathcal{R}_-, \Delta_{\max})$ that depends not on the true separation between two latent sources but instead on the minimum observed separation in the training data between two time series of opposite labels. In fact, our gap, in their setting, grows as $\Omega(\sigma^2 T)$ even when their gap $G^{(T)*}$ grows sublinear in $T$. In particular, while their result cannot handle the regime where $\mathcal{O}(\sigma^2 \log \frac{m}{\delta}) \leq G^{(T)*} \leq \sigma^2 \sqrt{T}$, ours can, using $n = \Theta(m \log \frac{m}{\delta})$ training time series and observing the first $T = \Omega(\log \frac{m}{\delta})$ time steps to classify a time series correctly with probability at least $1 - \delta$; see the longer version of this paper for details.

Vempala and Wang [17] have a spectral method for learning Gaussian mixture models that can handle smaller $G^{(T)*}$ than Dasgupta and Schulman's approach but requires $n = \widetilde{\Omega}(T^3 m^2)$ training data, where we've hidden the dependence on $\sigma^2$ and other variables of interest for clarity of presentation. Hsu and Kakade [10] have a moment-based estimator that doesn't have a gap condition but, under a different non-degeneracy condition, requires substantially more samples for our problem setup, i.e., $n = \Omega((m^{14} + Tm^{11})/\varepsilon^2)$ to achieve an $\varepsilon$ approximation of the mixture components. These results need substantially more training data than what we've shown is sufficient for classification.

To fit a Gaussian mixture model to massive training datasets, in practice, using all the training data could be prohibitively expensive. In such scenarios, one could instead non-uniformly subsample $\mathcal{O}(Tm^3/\varepsilon^2)$ time series from the training data using the procedure given in [8] and then feed the resulting smaller dataset, referred to as an $(m, \varepsilon)$-*coreset*, to the EM algorithm for learning the latent sources. This procedure still requires more training time series than needed for classification and lacks a guarantee that the estimated latent sources will be close to the true latent sources.

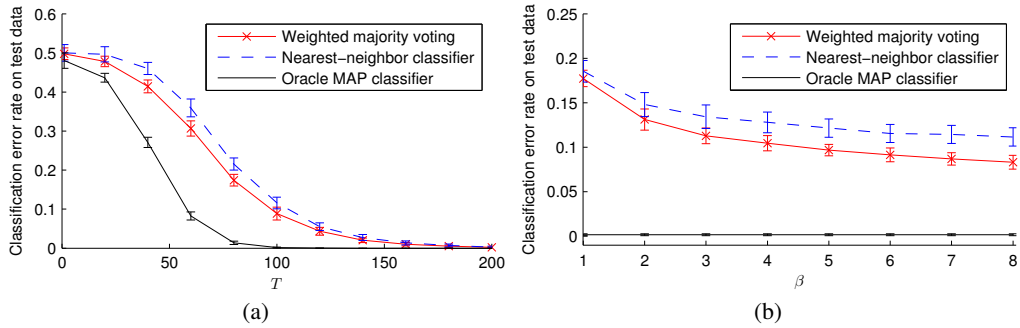

Figure 2: Results on synthetic data. (a) Classification error rate vs. number of initial time steps $T$ used; training set size: $n = \beta m \log m$ where $\beta = 8$. (b) Classification error rate at $T = 100$ vs. $\beta$. All experiments were repeated 20 times with newly generated latent sources, training data, and test data each time. Error bars denote one standard deviation above and below the mean value.

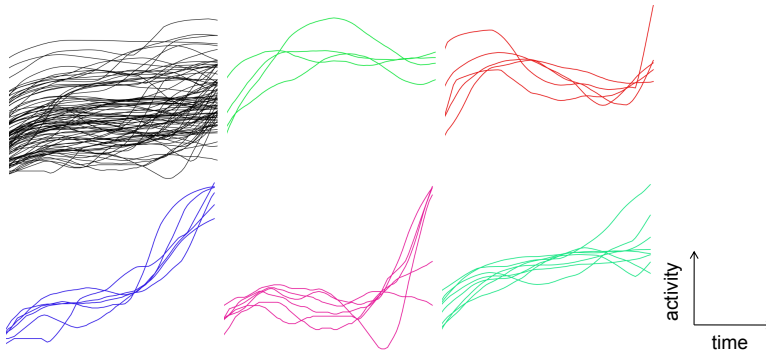

Figure 3: How news topics become trends on Twitter. The top left shows some time series of activity leading up to a news topic becoming trending. These time series superimposed look like clutter, but we can separate them into different clusters, as shown in the next five plots. Each cluster represents a "way" that a news topic becomes trending.

## 4 Experimental Results

**Synthetic data.** We generate $m = 200$ latent sources, where each latent source is constructed by first sampling i.i.d. $\mathcal{N}(0, 100)$ entries per time step and then applying a 1D Gaussian smoothing filter with scale parameter 30. Half of the latent sources are labeled $+1$ and the other half $-1$. Then $n = \beta m \log m$ training time series are sampled as per the latent source model where the noise added is i.i.d. $\mathcal{N}(0, 1)$ and $\Delta_{\max} = 100$. We similarly generate 1000 time series to use as test data. We set $\gamma = 1/8$ for weighted majority voting. For $\beta = 8$, we compare the classification error rates on test data for weighted majority voting, nearest-neighbor classification, and the MAP classifier with oracle access to the true latent sources as shown in Figure 2(a). We see that weighted majority voting outperforms nearest-neighbor classification but as $T$ grows large, the two methods' performances converge to that of the MAP classifier. Fixing $T = 100$, we then compare the classification error rates of the three methods using varying amounts of training data, as shown in Figure 2(b); the oracle MAP classifier is also shown but does not actually depend on training data. We see that as $\beta$ increases, both weighted majority voting and nearest-neighbor classification steadily improve in performance.

**Forecasting trending topics on twitter.** We provide only an overview of our Twitter results here, deferring full details to the longer version of this paper. We sampled 500 examples of trends at random from a list of June 2012 news trends, and 500 examples of non-trends based on phrases appearing in user posts during the same month. As we do not know how Twitter chooses what phrases are considered as candidate phrases for trending topics, it's unclear what the size of the

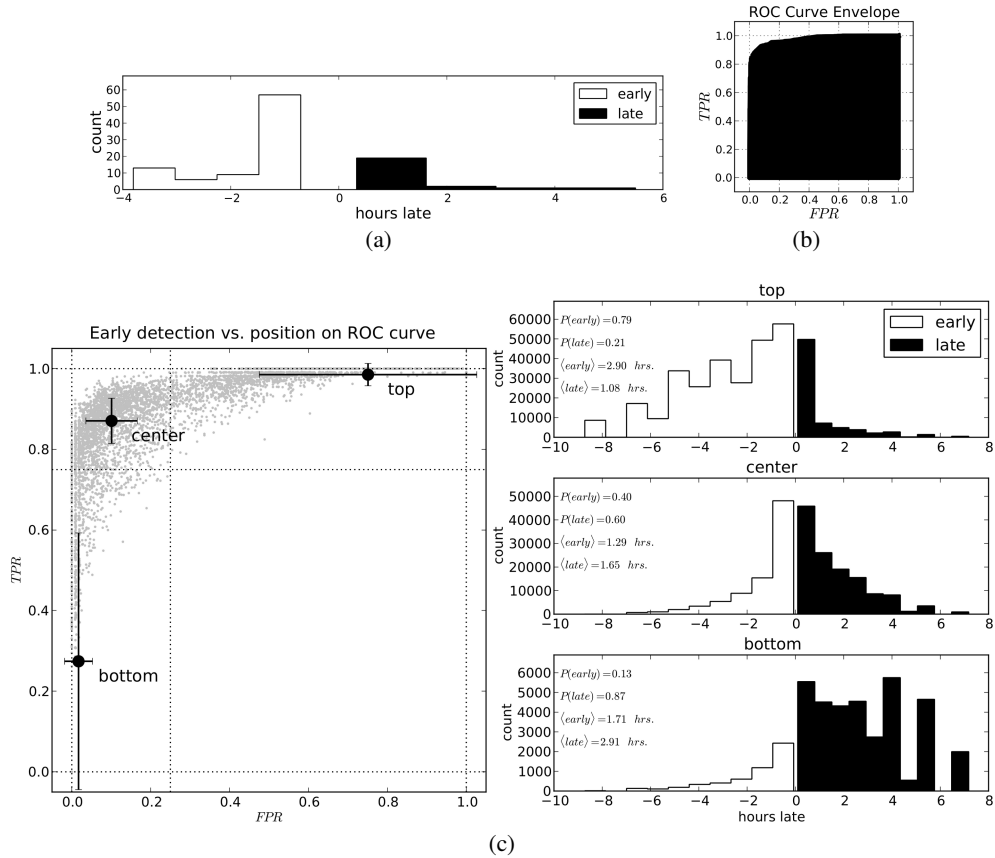

Figure 4: Results on Twitter data. (a) Weighted majority voting achieves a low error rate (FPR of 4%, TPR of 95%) and detects trending topics in advance of Twitter 79% of the time, with a mean of 1.43 hours when it does; parameters: $\gamma = 10, T = 115, T_{smooth} = 80, h = 7$. (b) Envelope of all ROC curves shows the tradeoff between TPR and FPR. (c) Distribution of detection times for "aggressive" (top), "conservative" (bottom) and "in-between" (center) parameter settings.

non-trend category is in comparison to the size of the trend category. Thus, for simplicity, we intentionally control for the class sizes by setting them equal. In practice, one could still expressly assemble the training data to have pre-specified class sizes and then tune $\theta$ for generalized weighted majority voting (8). In our experiments, we use the usual weighted majority voting (2) (i.e., $\theta = 1$) to classify time series, where $\Delta_{\max}$ is set to the maximum possible (we consider all shifts).

Per topic, we created its time series based on a pre-processed version of the raw rate of how often the topic was shared, i.e., its *Tweet rate*. We empirically found that how news topics become trends tends to follow a finite number of patterns; a few examples of these patterns are shown in Figure 3. We randomly divided the set of trends and non-trends into into two halves, one to use as training data and one to use as test data. We applied weighted majority voting, sweeping over $\gamma$, $T$, and data pre-processing parameters. As shown in Figure 4(a), one choice of parameters allows us to detect trending topics in advance of Twitter 79% of the time, and when we do, we detect them an average of 1.43 hours earlier. Furthermore, we achieve a true positive rate (TPR) of 95% and a false positive rate (FPR) of 4%. Naturally, there are tradeoffs between TPR, FPR, and how early we make a prediction (i.e., how small $T$ is). As shown in Figure 4(c), an "aggressive" parameter setting yields early detection and high TPR but high FPR, and a "conservative" parameter setting yields low FPR but late detection and low TPR. An "in-between" setting can strike the right balance.

**Acknowledgements.** This work was supported in part by the Army Research Office under MURI Award 58153-MA-MUR. GHC was supported by an NDSEG fellowship.

## Footnotes

[1]While it is not public knowledge how Twitter defines a topic to be a trending topic, Twitter does provide information for which topics are trending topics. We take these labels to be ground truth, effectively treating how a topic goes viral to be a black box supplied by Twitter.

[2]We index time using $\mathbb{Z}$ for notationally convenience but will assume time series to start at time step 1.

[3]While we keep the sampling uniform for clarity of presentation, our theoretical guarantees can easily be extended to the case where the sampling is not uniform. The only change is that the number of training data needed will be larger by a factor of $\frac{1}{m\pi_{\min}}$, where $\pi_{\min}$ is the smallest probability of a particular latent source occurring.

[4]We use a minimum rather a summation over time shifts to make the method more similar to existing time series classification work (e.g., [20]), which minimize over time warpings rather than simple shifts.

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
