[Supplementary Material]

# A  Proof of Theorem 1

Let $S$ be the time series with an unknown label that we wish to classify using training data. Denote $m_+ \triangleq |\mathcal{V}_+|$, $m_- \triangleq |\mathcal{V}_-| = m - m_+$, $n_+ \triangleq |\mathcal{R}_+|$, $n_- \triangleq |\mathcal{R}_-|$, and $\mathcal{R} \triangleq \mathcal{R}_+ \cup \mathcal{R}_-$. Recall that $\mathcal{D}_+ \triangleq \{0, 1, \ldots, \Delta_{\max}\}$, and $\mathcal{D} \triangleq \{-\Delta_{\max}, \ldots, -1, 0, 1, \ldots, \Delta_{\max}\}$.

As per the model, there exists a latent source $V$, shift $\Delta' \in \mathcal{D}_+$, and noise signal $E'$ such that

$$S = V * \Delta' + E'. \tag{15}$$

Applying a standard coupon collector's problem result, with a training set of size $n > \beta m \log m$, then with probability at least $1 - m^{-\beta+1}$, for each latent source $V \in \mathcal{V}$, there exists at least one time series $R$ in the set $\mathcal{R}$ of all training data that is generated from $V$. Henceforth, we assume that this event holds. In Appendix C, we elaborate on what happens if the latent sources are not uniformly sampled.

Note that $R$ is generated from $V$ as

$$R = V * \Delta'' + E'', \tag{16}$$

where $\Delta'' \in \mathcal{D}_+$ and $E''$ is a noise signal independent of $E'$. Therefore, we can rewrite $S$ in terms of $R$ as follows:

$$S = R * \Delta + E, \tag{17}$$

where $\Delta = \Delta' - \Delta'' \in \mathcal{D}$ (note the change from $\mathcal{D}_+$ to $\mathcal{D}$) and $E = E' - E'' * \Delta$. Since $E'$ and $E''$ are i.i.d. over time and sub-Gaussian with parameter $\sigma$, one can easily verify that $E$ is i.i.d. over time and sub-Gaussian with parameter $\sqrt{2}\sigma$.

We now bound the probability of error of classifier $\widehat{L}_\theta^{(T)}(\cdot; \gamma)$. The probability of error or misclassification using the first $T$ time steps of $S$ is given by

$$\mathbb{P}\big(\text{misclassify } S \text{ using its first } T \text{ time steps}\big)$$
$$= \mathbb{P}(\widehat{L}_\theta^{(T)}(S; \gamma) = -1 | L = +1) \underbrace{\mathbb{P}(L = +1)}_{m_+/m} + \mathbb{P}(\widehat{L}_\theta^{(T)}(S; \gamma) = +1 | L = -1) \underbrace{\mathbb{P}(L = -1)}_{m_-/m}. \tag{18}$$

In the remainder of the proof, we primarily show how to bound $\mathbb{P}(\widehat{L}_\theta^{(T)}(S; \gamma) = -1 | L = +1)$. The bound for $\mathbb{P}(\widehat{L}_\theta^{(T)}(S; \gamma) = +1 | L = -1)$ is almost identical. By Markov's inequality,

$$\mathbb{P}(\widehat{L}_\theta^{(T)}(S; \gamma) = -1 | L = +1) = \mathbb{P}\left(\frac{1}{\Lambda^{(T)}(S; \gamma)} \geq \frac{1}{\theta}\bigg| L = +1\right) \leq \theta \mathbb{E}\left[\frac{1}{\Lambda^{(T)}(S; \gamma)}\bigg| L = +1\right]. \tag{19}$$

Now,

$$\mathbb{E}\left[\frac{1}{\Lambda^{(T)}(S; \gamma)}\bigg| L = +1\right] \leq \max_{r_+ \in \mathcal{R}_+, \Delta_+ \in \mathcal{D}} \mathbb{E}_E\left[\frac{1}{\Lambda^{(T)}(r_+ * \Delta_+ + E; \gamma)}\right]. \tag{20}$$

With the above inequality in mind, we next bound $1/\Lambda^{(T)}(\widetilde{r}_+ * \widetilde{\Delta}_+ + E; \gamma)$ for any choice of $\widetilde{r}_+ \in \mathcal{R}_+$ and $\widetilde{\Delta}_+ \in \mathcal{D}$. Note that for any time series $s$,

$$\frac{1}{\Lambda^{(T)}(s; \gamma)} \leq \frac{\sum_{\substack{r_- \in \mathcal{R}_-, \\ \Delta_- \in \mathcal{D}}} \exp\big(-\gamma \|r_- * \Delta_- - s\|_T^2\big)}{\exp\big(-\gamma \|\widetilde{r}_+ * \widetilde{\Delta}_+ - s\|_T^2\big)}. \tag{21}$$

After evaluating the above for $s = \widetilde{r}_+ * \widetilde{\Delta}_+ + E$, a bit of algebra shows that

$$\frac{1}{\Lambda^{(T)}(\widetilde{r}_+ * \widetilde{\Delta}_+ + E; \gamma)}$$
$$\leq \sum_{\substack{r_- \in \mathcal{R}_-, \\ \Delta_- \in \mathcal{D}}} \big\{ \exp\big(-\gamma \|\widetilde{r}_+ * \widetilde{\Delta}_+ - r_- * \Delta_-\|_T^2\big) \exp\big(-2\gamma \langle \widetilde{r}_+ * \widetilde{\Delta}_+ - r_- * \Delta_-, E\rangle_T\big)\big\}, \tag{22}$$

where $\langle q, q'\rangle_T \triangleq \sum_{t=1}^T q(t)q'(t)$ for time series $q$ and $q'$.

Taking the expectation of (22) with respect to noise signal $E$, we obtain the following bound:

$$\mathbb{E}_E\left[\frac{1}{\Lambda^{(T)}(\widetilde{r}_+ * \widetilde{\Delta}_+ + E; \gamma)}\right]$$

$$\leq \mathbb{E}_E\left[\sum_{\substack{r_- \in \mathcal{R}_-, \\ \Delta_- \in \mathcal{D}}} \left\{\exp\left(-\gamma\|\widetilde{r}_+ * \widetilde{\Delta}_+ - r_- * \Delta_-\|_T^2\right)\exp\left(-2\gamma\langle\widetilde{r}_+ * \widetilde{\Delta}_+ - r_- * \Delta_-, E\rangle_T\right)\right\}\right]$$

$$\overset{(i)}{=} \sum_{\substack{r_- \in \mathcal{R}_-, \\ \Delta_- \in \mathcal{D}}} \exp\left(-\gamma\|\widetilde{r}_+ * \widetilde{\Delta}_+ - r_- * \Delta_-\|_T^2\right)\prod_{t=1}^{T}\mathbb{E}_{E(t)}[\exp\left(-2\gamma(\widetilde{r}_+(t + \widetilde{\Delta}_+) - r_-(t + \Delta_-))E(t)\right)]$$

$$\overset{(ii)}{\leq} \sum_{\substack{r_- \in \mathcal{R}_-, \\ \Delta_- \in \mathcal{D}}} \exp\left(-\gamma\|\widetilde{r}_+ * \widetilde{\Delta}_+ - r_- * \Delta_-\|_T^2\right)\prod_{t=1}^{T}\exp\left(4\sigma^2\gamma^2(\widetilde{r}_+(t + \widetilde{\Delta}_+) - r_-(t + \Delta_-))^2\right)$$

$$= \sum_{\substack{r_- \in \mathcal{R}_-, \\ \Delta_- \in \mathcal{D}}} \exp\left(-(\gamma - 4\sigma^2\gamma^2)\|r_+ * \Delta_+ - r_- * \Delta_-\|_T^2\right)$$

$$\leq (2\Delta_{\max} + 1)n_- \exp\left(-(\gamma - 4\sigma^2\gamma^2)G^{(T)}\right), \tag{23}$$

where step $(i)$ uses independence of entries of $E$, step $(ii)$ uses the fact that $E(t)$ is zero-mean sub-Gaussian with parameter $\sqrt{2}\sigma$, and the last line abbreviates the gap $G^{(T)} \equiv G^{(T)}(\mathcal{R}_+, \mathcal{R}_-, \Delta_{\max})$.

Stringing together inequalities (19), (20), and (23), we obtain

$$\mathbb{P}(\widehat{L}_\theta^{(T)}(S; \gamma) = -1|L = +1) \leq \theta(2\Delta_{\max} + 1)n_- \exp\left(-(\gamma - 4\sigma^2\gamma^2)G^{(T)}\right). \tag{24}$$

Repeating a similar argument yields

$$\mathbb{P}(\widehat{L}_\theta^{(T)}(S; \gamma) = +1|L = -1) \leq \frac{1}{\theta}(2\Delta_{\max} + 1)n_+ \exp\left(-(\gamma - 4\sigma^2\gamma^2)G^{(T)}\right). \tag{25}$$

Finally, plugging (24) and (25) into (18) gives

$$\mathbb{P}(\widehat{L}_\theta^{(T)}(S; \gamma) \neq L) \leq \theta(2\Delta_{\max} + 1)\frac{n_- m_+}{m} \exp\left(-(\gamma - 4\sigma^2\gamma^2)G^{(T)}\right)$$

$$+ \frac{1}{\theta}(2\Delta_{\max} + 1)\frac{n_+ m_-}{m}(2\Delta_{\max} + 1)n_+ \exp\left(-(\gamma - 4\sigma^2\gamma^2)G^{(T)}\right)$$

$$= \left(\frac{\theta m_+}{m} + \frac{m_-}{\theta m}\right)(2\Delta_{\max} + 1)n \exp\left(-(\gamma - 4\sigma^2\gamma^2)G^{(T)}\right). \tag{26}$$

This completes the proof of Theorem 1.

## B  Proof of Theorem 2

The proof uses similar steps as the weighted majority voting case. As before, we consider the case when our training data sees each latent source at least once (this event happens with probability at least $1 - m^{-\beta+1}$).

We decompose the probability of error into terms depending on which latent source $V$ generated $S$:

$$\mathbb{P}(\widehat{L}_{NN}^{(T)}(S) \neq L) = \sum_{v \in \mathcal{V}}\mathbb{P}(V = v)\mathbb{P}(\widehat{L}_{NN}^{(T)}(S) \neq L|V = v) = \sum_{v \in \mathcal{V}}\frac{1}{m}\mathbb{P}(\widehat{L}_{NN}^{(T)}(S) \neq L|V = v). \tag{27}$$

Next, we bound each $\mathbb{P}(\widehat{L}_{NN}^{(T)}(S) \neq L|V = v)$ term. Suppose that $v \in \mathcal{V}_+$, i.e., $v$ has label $L = +1$; the case when $v \in \mathcal{V}_-$ is similar. Then we make an error and declare $\widehat{L}_{NN}^{(T)}(S) = -1$ when the nearest neighbor $\widehat{r}$ to time series $S$ is in the set $\mathcal{R}_-$, where

$$(\widehat{r}, \widehat{\Delta}) = \arg\min_{(r,\Delta) \in \mathcal{R} \times \mathcal{D}}\|r * \Delta - S\|_T^2. \tag{28}$$

By our assumption that every latent source is seen in the training data, there exists $r^* \in \mathcal{R}_+$ generated by latent source $v$, and so

$$S = r^* * \Delta^* + E \tag{29}$$

for some shift $\Delta^* \in \mathcal{D}$ and noise signal $E$ consisting of i.i.d. entries that are zero-mean sub-Gaussian with parameter $\sqrt{2}\sigma$.

By optimality of $(\widehat{r}, \widehat{\Delta})$ for optimization problem (28), we have

$$\|r * \Delta - (r^* * \Delta^* + E)\|_T^2 \geq \|\widehat{r} * \widehat{\Delta} - (r^* * \Delta^* + E)\|_T^2 \qquad \text{for all } r \in \mathcal{R}, \Delta \in \mathcal{D}. \tag{30}$$

Plugging in $r = r^*$ and $\Delta = \Delta^*$, we obtain

$$\begin{aligned}
\|E\|_T^2 &\geq \|\widehat{r} * \widehat{\Delta} - (r^* * \Delta^* + E)\|_T^2 \\
&= \|(\widehat{r} * \widehat{\Delta} - r^* * \Delta^*) - E\|_T^2 \\
&= \|\widehat{r} * \widehat{\Delta} - r^* * \Delta^*\|_T^2 - 2\langle \widehat{r} * \widehat{\Delta} - r^* * \Delta^*, E\rangle_T + \|E\|_T^2,
\end{aligned} \tag{31}$$

or, equivalently,

$$2\langle \widehat{r} * \widehat{\Delta} - r^* * \Delta^*, E\rangle_T \geq \|\widehat{r} * \widehat{\Delta} - r^* * \Delta^*\|_T^2. \tag{32}$$

Thus, given $V = v \in \mathcal{V}_+$, declaring $\widehat{L}_{NN}^{(T)}(S) = -1$ implies the existence of $\widehat{r} \in \mathcal{R}_-$ and $\widehat{\Delta} \in \mathcal{D}$ such that optimality condition (32) holds. Therefore,

$$\begin{aligned}
&\mathbb{P}(\widehat{L}_{NN}^{(T)}(S) = -1 | V = v) \\
&\leq \mathbb{P}\left( \bigcup_{\widehat{r} \in \mathcal{R}_-, \widehat{\Delta} \in \mathcal{D}} \{2\langle \widehat{r} * \widehat{\Delta} - r^* * \Delta^*, E\rangle_T \geq \|\widehat{r} * \widehat{\Delta} - r^* * \Delta^*\|_T^2\} \right) \\
&\overset{(i)}{\leq} (2\Delta_{\max} + 1)n_- \mathbb{P}(2\langle \widehat{r} * \widehat{\Delta} - r^* * \Delta^*, E\rangle_T \geq \|\widehat{r} * \widehat{\Delta} - r^* * \Delta^*\|_T^2) \\
&\leq (2\Delta_{\max} + 1)n_- \mathbb{P}(\exp(2\lambda\langle \widehat{r} * \widehat{\Delta} - r^* * \Delta^*, E\rangle_T) \geq \exp(\lambda\|\widehat{r} * \widehat{\Delta} - r^* * \Delta^*\|_T^2)) \\
&\overset{(ii)}{\leq} (2\Delta_{\max} + 1)n_- \exp(-\lambda\|\widehat{r} * \widehat{\Delta} - r^* * \Delta^*\|_T^2)\mathbb{E}[\exp(2\lambda\langle \widehat{r} * \widehat{\Delta} - r^* * \Delta^*, E\rangle_T)] \\
&\overset{(iii)}{\leq} (2\Delta_{\max} + 1)n_- \exp(-\lambda\|\widehat{r} * \widehat{\Delta} - r^* * \Delta^*\|_T^2) \exp(4\lambda^2\sigma^2\|\widehat{r} * \widehat{\Delta} - r^* * \Delta^*\|_T^2) \\
&= (2\Delta_{\max} + 1)n_- \exp(-(\lambda - 4\lambda^2\sigma^2)\|\widehat{r} * \widehat{\Delta} - r^* * \Delta^*\|_T^2) \\
&\leq (2\Delta_{\max} + 1)n \exp(-(\lambda - 4\lambda^2\sigma^2)G^{(T)}) \\
&\overset{(iv)}{\leq} (2\Delta_{\max} + 1)n \exp\left( -\frac{1}{16\sigma^2}G^{(T)} \right),
\end{aligned} \tag{33}$$

where step $(i)$ is by a union bound, step $(ii)$ is by Markov's inequality, step $(iii)$ is by sub-Gaussianity, and step $(iv)$ is by choosing $\lambda = \frac{1}{8\sigma^2}$.

As bound (33) also holds for $\mathbb{P}(\widehat{L}_{NN}^{(T)}(S) = +1 | V = v)$ when instead $v \in \mathcal{V}_-$, we can now piece together (27) and (33) to yield the final result:

$$\mathbb{P}(\widehat{L}_{NN}^{(T)}(S) \neq L) = \sum_{v \in \mathcal{V}} \frac{1}{m}\mathbb{P}(\widehat{L}_{NN}^{(T)}(S) \neq L | V = v) \leq (2\Delta_{\max} + 1)n \exp\left( -\frac{1}{16\sigma^2}G^{(T)} \right). \tag{34}$$

## C    Handling Non-uniformly Sampled Latent Sources

When each time series generated from the latent source model is sampled uniformly at random, then having $n > m\log\frac{2m}{\delta}$ (i.e., $\beta = 1 + \log\frac{2}{\delta}/\log m$) ensures that with probability at least $1 - \frac{\delta}{2}$, our training data sees every latent source at least once. When the latent sources aren't sampled uniformly at random, we show that we can simply replace the condition $n > m\log\frac{2m}{\delta}$ with $n \geq \frac{8}{\pi_{\min}}\log\frac{2m}{\delta}$ to achieve a similar (in fact, stronger) guarantee, where $\pi_{\min}$ is the smallest probability of a particular latent source occurring.

**Lemma 3.** *Suppose that the $i$-th latent source occurs with probability $\pi_i$ in the latent source model. Denote $\pi_{\min} \triangleq \min_{i \in \{1,2,\dots,m\}} \pi_i$. Let $\xi_i$ be the number of times that the $i$-th latent source appears in the training data. If $n \geq \frac{8}{\pi_{\min}} \log \frac{2m}{\delta}$, then with probability at least $1 - \frac{\delta}{2}$, every latent source appears strictly greater than $\frac{1}{2} n \pi_{\min}$ times in the training data.*

*Proof.* Note that $\xi_i \sim \mathrm{Bin}(n, \pi_i)$. We have

$$
\begin{aligned}
\mathbb{P}\big(\xi_i \leq \tfrac{1}{2} n \pi_{\min}\big) &\leq \mathbb{P}\big(\xi_i \leq \tfrac{1}{2} n \pi_i\big) \\
&\overset{(i)}{\leq} \exp\Big( -\frac{1}{2} \cdot \frac{(n\pi_i - \frac{1}{2} n\pi_i)^2}{n \cdot \pi_i} \Big) \\
&= \exp\Big( -\frac{n\pi_i}{8} \Big) \\
&\leq \exp\Big( -\frac{n\pi_{\min}}{8} \Big).
\end{aligned}
\tag{35}
$$

where step $(i)$ uses a standard binomial distribution lower tail bound. Applying a union bound,

$$
\mathbb{P}\Big( \bigcup_{i \in \{1,2,\dots,m\}} \big\{\xi_i \leq \tfrac{1}{2} n \pi_{\min}\big\} \Big) \leq m \exp\Big( -\frac{n\pi_{\min}}{8} \Big),
\tag{36}
$$

which is at most $\frac{\delta}{2}$ when $n \geq \frac{8}{\pi_{\min}} \log \frac{2m}{\delta}$. $\qquad\square$

## D  Forecasting Trending Topics on Twitter

Twitter is a social network whose users post messages called *Tweets*, which are then broadcast to a user's followers. Often, emerging topics of interest are discussed on Twitter in real time. Inevitably, certain topics gain sudden popularity and — in Twitter speak — begin to *trend*. Twitter surfaces such topics as a list of top ten *trending topics*, or *trends*.

**Data.** We sampled 500 examples of trends at random from a list of June 2012 news trends and recorded the earliest time each topic trended within the month. Before sampling, we filtered out trends that never achieved a rank of 3 or better on the Twitter trends list[5] as well as trends that lasted for less than 30 minutes as to keep our trend examples reasonably salient. We also sampled 500 examples of non-trends at random from a list of $n$-grams (of sizes 1, 2, and 3) appearing in Tweets created in June 2012, where we filter out any $n$-gram containing words that appeared in one of our 500 chosen trend examples. Note that as we do not know how Twitter chooses what phrases are considered as topic phrases (and are candidates for trending topics), it's unclear what the size of the non-trend category is in comparison to the size of the trend category. Thus, for simplicity, we intentionally control for the class sizes by setting them equal. In practice, one could still expressly assemble the training data to have pre-specified class sizes and then tune $\theta$ for generalized weighted majority voting (8). In our experiments, we just use the usual weighted majority voting (2) (i.e., $\theta = 1$) to classify time series.

From these examples of trends and non-trends, we then created time series of activity for each topic based on the rate of Tweets about that topic over time. To approximate this rate, we gathered 10% of all Tweets from June 2012, placed them into two-minute buckets according to their timestamps, and counted the number of Tweets in each bucket. We denote the count at the $t$-th time bucket as $\rho(t)$, which we refer to as the raw rate. We then transform the raw rate in a number of ways, summarized in Figure 5, before using the resulting time series for classification.

We observed that trending activity is characterized by spikes above some baseline rate, whereas non-trending activity has fewer, if any spikes. For example, a non-trending topic such as "city" has a very high, but mostly constant rate because it is a common word. In contrast, soon-to-be-trending topics like "Miss USA" will initially have a low rate, but will also have bursts in activity as the news spreads. To emphasize the parts of the rate signal above the baseline and de-emphasize the parts below the baseline, we define a baseline-normalized signal $\rho_b(t) \triangleq \rho(t) / \sum_{\tau=1}^{t} \rho(\tau)$.

$\rho(t) \rightarrow$ [Baseline normalization] $\xrightarrow{\rho_b(t)}$ [Emphasize large spikes] $\xrightarrow{\rho_{b,s}(t)}$ [Smoothing] $\xrightarrow{\rho_{b,s,c}(t)}$ [Log] $\rightarrow \rho_{b,s,c,l}(t)$

Figure 5: Twitter data pre-processing pipeline: The raw rate $\rho(t)$ counts the number of Tweets in time bucket $t$. We normalize $\rho(t)$ to make the counts relative: $\rho_b(t) \triangleq \rho(t)/\sum_{\tau=1}^{t} \rho(\tau)$. Large spikes are emphasized: $\rho_{b,s}(t) \triangleq |\rho_b(t) - \rho_b(t-1)|^\alpha$ (we use $\alpha = 1.2$). Next, we smooth the signal: $\rho_{b,s,c}(t) \triangleq \sum_{\tau=t-T_{smooth}+1}^{t} \rho_{b,s}(\tau)$. Finally, we take the log: $\rho_{b,s,c,l}(t) \triangleq \log \rho_{b,s,c}(t)$.

A related observation is that the Tweet rate for a trending topic typically contains larger and more sudden spikes than those of non-trending topics. We reward such spikes by emphasizing them, while de-emphasizing smaller spikes. To do so, we define a baseline-and-spike-normalized rate $\rho_{b,s}(t) \triangleq |\rho_b(t) - \rho_b(t-1)|^\alpha$ in terms of the already baseline-normalized rate $\rho_b$; parameter $\alpha \geq 1$ controls how much spikes are rewarded (we used $\alpha = 1.2$). In addition, we convolve the result with a smoothing window to eliminate noise and effectively measure the volume of Tweets in a sliding window of length $T_{smooth}$: $\rho_{b,s,c}(t) \triangleq \sum_{\tau=t-T_{smooth}+1}^{t} \rho_{b,s}(\tau)$.

Finally, the spread of a topic from person to person can be thought of as a branching process in which a population of users "affected" by a topic grows exponentially with time, with the exponent depending on the details of the model [21]. This intuition suggests using a logarithmic scaling for the volume of Tweets: $\rho_{b,s,c,l}(t) \triangleq \log \rho_{b,s,c}(t)$.

The resulting time series $\rho_{b,s,c,l}$ contains data from the entire window in which data was collected. To construct the sets of training time series $\mathcal{R}_+$ and $\mathcal{R}_-$, we keep only a small $h$-hour slice of representative activity $r$ for each topic. Namely, each of the final time series $r$ used in the training data is truncated to only contain the $h$ hours of activity in the corresponding transformed time series $\rho_{b,s,c,l}$. For time series corresponding to trending topics, these $h$ hours are taken from the time leading up to when the topic was first declared by Twitter to be trending. For time series corresponding to non-trending topics, the $h$-hour window of activity is sampled at random from all the activity for the topic. We empirically found that how news topics become trends tends to follow a finite number of patterns; a few examples of these patterns are shown in Figure 3.

**Experiment.** For a fixed choice of parameters, we randomly divided the set of trends and non-trends into two halves, one for training and one for testing. Weighted majority voting with the training data was used to classify the test data. Per time series in the test data, we looked within a window of $2h$ hours, centered at the trend onset for trends, and sampled randomly for non-trends. We restrict detection to this time window to avoid detecting earlier times that a topic became trending, if it trended multiple times. We then measured the false positive rate (FPR), true positive rate (TPR), and the time of detection if any. For trends, we computed how early or late the detection was compared to the true trend onset. We explored the following parameters: $h$, the length in hours of each example time series; $T$, the number of initial time steps in the observed time series $s$ that we use for classification; $\gamma$, the scaling parameter; $T_{smooth}$, the width of the smoothing window. In all cases, constant $\Delta_{\max}$ in the decision rule (2) is set to be the maximum possible, i.e., since observed signal $s$ has $T$ samples, we compare $s$ with all $T$-sized chunks of each time series $r$ in training data.

For a variety of parameters, we detect trending topics before they appear on Twitter's trending topics list. Figure 4 (a) shows that for one such choice of parameters, we detect trending topics before Twitter does 79% of the time, and when we do, we detect them an average of 1.43 hours earlier. Furthermore, we achieve a TPR of 95% and a FPR of 4%. Naturally, there are tradeoffs between the FPR, the TPR, and relative detection time that depend on parameter settings. An aggressive parameter setting will yield early detection and a high TPR, but at the expense of a high FPR. A conservative parameter setting will yield a low FPR, but at the expense of late detection and a low TPR. An in-between setting can strike the right balance. We show this tradeoff in two ways. First, by varying a single parameter at a time and fixing the rest, we generated an ROC curve that describes the tradeoff between FPR and TPR. Figure 4 (b) shows the envelope of all ROC curves, which can be interpreted as the best "achievable" ROC curve. Second, we broke the results up by where they fall on the ROC curve — top ("aggressive"), bottom ("conservative"), and center ("in-between") — and showed the distribution of early and late relative detection times for each (Figure 4(c)).

We discuss some fine details of the experimental setup. Due to restrictions on the Twitter data available, while we could determine whether a trending topic is categorized as news based on user-curated lists of "news" people on Twitter, we did not have such labels for individual Tweets. Thus, the example time series that we use as training data contain Tweets that are both news and non-news. We also reran our experiments using only non-news Tweets and found similar results except that we do not detect trends as early as before; however, weighted majority voting still detects trends in advance of Twitter 79% of the time.

## Footnotes

[5] On Twitter, trending topics compete for the top ten spots whereas we are only detecting whether a topic will trend or not.