[Reviews · NeurIPS 2013]

Submitted by Assigned_Reviewer_4

The authors examine theoretically and empirically properties of the k=1-nearest neighbor classification (NN) and a trivial variant of the kernel density classification, called “weighted majority voting”, in a time-series binary classification problem. For theoretical analysis, a simplified generative model for time-series is introduced. Under this model, they provide non-asymptotic performance guarantees in terms of how large of a training dataset and how much of the time-series length to be classified.

Although the theoretical analysis is technically sound, the empirical work will be incomplete. In the experiments, the weighted majority voting and the NN are just tested while data analysts would at least use the k-NN, rather than the NN, with a tuned ‘k’ by cross-validation or so. I think the authors should apply k-NN as a baseline. Otherwise, I feel that they evaluate the presented results only from advantageous viewpoint. Also there exist several methods [i, ii, iii] similar to the (proposed) weighted majority voting.

The linkage between theoretical and empirical parts seems to be week. While the presented upper error-bound of the NN seems to be always lower than or equal to that of the weighted majority voting, the weighted majority voting outperformed the NN in the experiments. Some careful discussion about this will be needed for the final version.

Minors:
- It would be better to explain why the summations over time shifts need to be replaced by the minimums in Eq. (7).
- To understand the claims such in lines 284~286, where the constant factor of (\theta + 1/\theta) in Eq. (10) can be replace by 1, Remark in the supplementary material is essential. The authors should introduce the content in the remark in the main paper.
- It would be helpful for readers if the bounds of Eq (10) and (13) are plotted with respected to G with various settings of \gamma and \sigma.
- There is no conclusion or summary section.

[i] Sahibsingh A. Dudani: The distance-weighted k-nearest neighbor rule, IEEE Transactions on System, Man, and Cybernetics, 6 (1976), 325-327.
[ii]Thanh N. Tran, Ron Wehrens, Lutgarde M. C. Buydens: KNN-kernel density-based clustering for high-dimensional multivariate data. Computational Statistics & Data Analysis 51(2): 513-525 (2006).
[iii] Jianping Gou, Taisong Xiong, Yin Kuang: A Novel Weighted Voting for K-Nearest Neighbor Rule. JCP 6(5): 833-840 (2011)
Summary: The presented model and algorithms seem to be too simple. Nevertheless, the paper addresses difficult, important questions in the time-series classification and provides theoretical guarantees for the algorithms under the model.

Submitted by Assigned_Reviewer_5

This paper presents theoretical results on the efficacy of nearest neighbor based methods to the problem of online and offline time series classification. They assume that data is sampled from one of many latent sources with a zero mean sub-Gaussian additive noise model. Each source is assumed to be associated with one of the class labels. To classify individual series, they show that estimating the latent sources are not necessary (and inefficient). Instead, they evaluate for a weighted majority vote and a nearest neighbor based algorithm. They characterize separability of the data as a function of the minimum gap between two series in the two opposite classes. For the weighted majority voting algorithm to classify correctly with high probability, they give conditions on how the gap must grow as a function of the number of latent sources. When the latent sources are separable, and an additional criterion about the gap is met (namely, nearest two training series from opposite classes are not within noise of each other), they also show that observing the first \omega(# of latent sources/ delta) points within a series is sufficient to classify the series with probability at least 1-\delta.

This is a well-written paper. The results are practically relevant in devising algorithms for time-series classification. They show results on classifying Twitter topics for whether they will trend. Using the majority weighted vote, they are able to detect topics that will trend more than an hour sooner than Twitter’s blackbox algorithm.

Questions and comments
1. Pg 4 Ln 177 Justify here why zero-mean sub-Gaussian?
2. Can you comment within the paper on how your analysis depends on the way the distance metric (and gap) is defined.
Summary: This paper presents theoretical results on the efficacy of nearest neighbor based methods to the problem of online and offline time series classification. This is a well-written paper. The results are practically relevant in devising algorithms for time-series classification. They also show empirical results on classifying Twitter topics for whether they will trend. Using the majority weighted vote, they are able to detect topics that will trend more than an hour sooner than Twitter’s blackbox algorithm.

Submitted by Assigned_Reviewer_6

[I have not read the supplemental material]

This paper introduces a latent source model for time-series classification and use it to derive bounds on the performance of weighted majority voting and k-nearest neighbor. The weighted majority voting classifier is then used to predict whether twitter topics will become "trending topics", and is shown to predict twitter's own gold labels reliably about an hour earlier.


Major Comments

This paper makes an interesting and novel contribution to non-parametric time-series classification. The intuition behind the latent source model is clear, and I believe encodes reasonable assumptions about real-world time-series might behave.

My main concern is that I don't have a good conception of where the latent source model might break down—are there pathological real-world cases that cannot be easily fit into this type of model? What are the limitations of the scope of the results here?

Minor Comments:

-- (4.1): I would like to understand figure 3 better. Are there semantic similarities in the latent trending topic groups? I.e. can you say something about the content of topics that trend one way vs trend another? Or barring that is the explanation due to network effects? In general placing this result in context with other twitter work on trending topics.

-- (4.1): Significance of the differences in Figure 2?

-- (4.2): How much would the results change if you tried to do the true, unbalanced class prediction problem? That is, say I want to predict trending topics _right now_ then I won't have nice balanced classes like in the eval setup. Instead, only a tiny fraction of my time series will ever start trending. How well does the weighted-majority approach handle these tail cases?
Summary: This paper makes an interesting and novel contribution to non-parametric time-series classification. The intuition behind the latent source model is clear, and I believe encodes reasonable assumptions about real-world time-series might behave.
Author Feedback

Author rebuttal: We thank all the reviewers for detailed feedback and appreciate their time and effort.

Our work focuses on developing theoretical understanding of non-parametric, nearest-neighbor-like approaches for time series classification. To this end, our empirical results with synthetic data chiefly serve to elucidate our theoretical results for 1NN classification and weighted majority voting; hence, we have not provided comparisons against other methods (e.g., kNN classification for k>1). Meanwhile, our empirical results with real data on forecasting Twitter trends provides strong support of the validity of our model in practice. We remark that since a kNN classifier (weighted as well as unweighted) is very similar to what we present, we believe that our proof ideas could extend to showing when kNN classification should also work well under the latent source model; additional assumptions may be needed to prove performance guarantees in terms of k.

Regarding the latent source model itself, indeed the noise model we use right now is rather simplistic as to enable the analysis. It would be interesting to see if the analysis could extend to a more complicated noise model. In any case, the sub-Gaussian setting presented essentially allows the theory to work a lot like the Gaussian case, hence why the gap defined is squared Euclidean distance. Also, the reason why we take the minimum over shifts rather than just use the sum is to make the method more similar to existing time series classification work, which minimize over dynamic time warpings rather than simple shifts.

As for how the model might break down, unfortunately, we currently do not have a good handle on how the algorithms degrade when we move away from the model, nor do we have a concrete test for whether a dataset fits the model well. This remains an interesting future direction.

In terms of the comparison between 1NN classification and weighted majority voting, we remark that the MAP classifier that knows the latent sources would actually always choose theta=1 (put another way, we don't weight false positives and false negatives differently), and with theta=1, our bounds for the two different algorithms actually matches with the appropriate choice of gamma. This is not surprising since the proof technique used for both is essentially the same--we analyze the event where only 1 good training time series is found (with the same latent source as the observation), and that there are n bad training time series (all with the latent source as far away from the true latent source and that has the wrong label). As for the empirical results, while we did find weighted majority voting to almost always do better than 1NN classification for small T, it would indeed be worthwhile to check the significance of the differences.